# Proteomic Identification and Meta-Analysis in *Salvia hispanica* RNA-Seq de novo Assemblies

**DOI:** 10.3390/plants10040765

**Published:** 2021-04-14

**Authors:** Ashwil Klein, Lizex H. H. Husselmann, Achmat Williams, Liam Bell, Bret Cooper, Brent Ragar, David L. Tabb

**Affiliations:** 1Department of Biotechnology, University of the Western Cape, Bellville 7535, South Africa; aklein@uwc.ac.za (A.K.); lhusselmann@uwc.ac.za (L.H.H.H.); Achmatwilliams04@gmail.com (A.W.); 2Centre for Proteomic and Genomic Research, Cape Town 7925, South Africa; Liam.Bell@cpgr.org.za; 3USDA Agricultural Research Service, Beltsville, MD 20705, USA; bret.cooper@usda.gov; 4Departments of Internal Medicine and Pediatrics, Massachusetts General Hospital, Harvard Medical School, Boston, MA 02150, USA; bragar@mgh.harvard.edu; 5Division of Molecular Biology and Human Genetics, Faculty of Medicine and Health Sciences, Stellenbosch University, Cape Town 7500, South Africa; 6Centre for Bioinformatics and Computational Biology, Stellenbosch University, Stellenbosch 7602, South Africa

**Keywords:** proteogenomics, *Salvia hispanica*, *Salvia columbariae*, nonmodel organisms, LC-MS/MS, bioinformatics, RNA-Seq, proteomics

## Abstract

While proteomics has demonstrated its value for model organisms and for organisms with mature genome sequence annotations, proteomics has been of less value in nonmodel organisms that are unaccompanied by genome sequence annotations. This project sought to determine the value of RNA-Seq experiments as a basis for establishing a set of protein sequences to represent a nonmodel organism, in this case, the pseudocereal chia. Assembling four publicly available chia RNA-Seq datasets produced transcript sequence sets with a high BUSCO completeness, though the number of transcript sequences and Trinity “genes” varied considerably among them. After six-frame translation, ProteinOrtho detected substantial numbers of orthologs among other species within the taxonomic order *Lamiales*. These protein sequence databases demonstrated a good identification efficiency for three different LC-MS/MS proteomics experiments, though a seed proteome showed considerable variability in the identification of peptides based on seed protein sequence inclusion. If a proteomics experiment emphasizes a particular tissue, an RNA-Seq experiment incorporating that same tissue is more likely to support a database search identification of that proteome.

## 1. Introduction

Pseudocereals are broad-leaf nongrass species that produce fruits and/or seeds similar to conventional cereal crops with considerable nutritive value [1]. Although there are several plants under this group, the four that have garnered the most attention include quinoa, amaranth, buckwheat, and chia.

Chia (*Salvia hispanica L*.) is a pseudocereal crop indigenous to Mesoamerican countries, including Mexico and Guatemala. The cultivation of chia has recently expanded to nations around the world because the plant seeds are rich in dietary nutrients (fiber, antioxidants, minerals and vitamins, proteins, and unsaturated fatty acids) and phenolic compounds often associated with health benefits [2]. Apart from the nutritional value of the seeds, chia leaves and stems are a good source of polyunsaturated fatty acids (PUFA) and other essential oils, which are used as feed additives in ruminant nutrition [3,4].

In recent years, chia has received considerable attention due to its nutritive value and medicinal benefit in various chronic and lifestyle diseases. A few lines of research have shown that chia seed consumption is beneficial in regulating symptoms associated with inflammation, cardiovascular diseases, and insulin resistance [5,6,7]. The high levels of omega-3 fatty acid content in chia seeds may help reduce the risk of cardiovascular disease and stroke. Omega-3 fatty acids, not naturally produced by the human body, are critical for maintaining healthy blood pressure and cholesterol, as well as reducing inflammation and coagulation [8]. Although it is hypothesized that the consumption of chia seeds may be beneficial to human health, published human trials have been small and short, and they have assessed the secondary markers of health rather than primary outcomes [9].

Since the popularization of liquid chromatography-tandem mass spectrometry (LC-MS/MS) [10], a technology to identify thousands of peptides and, thus, proteins in complex mixtures, proteomics has become broadly accessible and highly sensitive, enabling clinical research and biological discoveries. The standard algorithm for matching peptide tandem mass spectra to peptide sequences is the database search algorithm [11], which requires as an input the list of all protein sequences that may be represented in a sample. For model organisms with mature genome sequence annotations, such as *Arabidopsis thaliana*, the full complement of protein sequences can be predicted with good accuracy [12]. If a genome sequence annotation is unavailable, however, protein sequences have generally been based upon cDNA libraries, which frequently omit substantial numbers of proteins. In a recent example, a bean proteome identified 1400 proteins using a cDNA library of 9000 sequences, but the same data identified 4000 proteins when an annotated genome sequence became available [13,14].

A key feature of database search engines is that they typically expect the observed masses of peptide ions to match the calculated masses of peptide sequences to a high degree of mass accuracy, for example, 20 ppm (for a 2000 Da peptide, 20 ppm is a mass error of 0.04 Da). If the sequence database contains only an approximate match for the proteins being observed (such as a substitution of Leu by Val in the protein sequence), that sequence disparity would cause a mass difference large enough that the correct sequence would not be compared to this tandem mass spectrum.

The first forays of proteomics into chia featured *Salvia hispanica* proper (2010) [15], *S. miltiorrhiza* medicinal herbs (2010) [16], and *S. splendens* ornamental plants (2013) [17]. While the latter two studies employed 2D gels and MALDI-TOF/TOF from gel spots, the lack of sequence data limited the number of identified proteins to a handful. The pioneering work in *S. hispanica* by Olivos-Lugo et al. quantified amino acids by a Beckman Instruments System 6300 rather than a mass spectrometer.

The primary question that motivated this study was this: “Are transcript sequences assembled from RNA-Seq experiments sufficiently complete to allow for proteome identification?” This study published two new LC-MS/MS data sets for *Salvia spp*., employing Thermo Orbitrap Fusion Lumos and Q Exactive instruments. The authors of recently published seed proteomics papers [18,19] have graciously agreed to share their Q Exactive Plus data, as well. We have analyzed these peptide tandem mass spectra using protein information derived from four contemporary RNA-Seq experiments, taking care to document each bioinformatics step required. Overall, we demonstrate that protein identification based on assembled transcript sequence sets is feasible even in the absence of an annotated genome sequence, allowing the identification of tens of thousands of peptides from multiple plant tissues.

## 2. Results

Similar to many commercially important crops, chia has benefited from the ubiquitous application of massively parallel sequencing. Perhaps unusually, many of the projects included in the NCBI Sequence Read Archive for this species emphasize paired-end RNA-Seq rather than whole genome shotgun experiments. We decided on a meta-analysis spanning four RNA-Seq projects: PRJNA196477 (Sreedhar) [20], PRJNA448759 (Peláez) [21], PRJNA597830 (Wimberley) [22], and PRJEB19614 (Gupta) [23]. Table 1 compares these experiments. The Peláez data were further subdivided to four strains: The spotted cultivars grown in Celaya and Veracruz were analyzed in triplicate experiments while the Cualac cultivar grown in Celaya and the spotted cultivar grown in Jalisco were analyzed in duplicates.

### 2.1. Trimming and Assembly

The Harvard FAS Informatics and Scientific Applications group best practices guide describes two orthogonal “trimming” challenges for assembling transcript sequences from RNA-Seq data. The first, removing reads or truncating reads on the basis of low base-calling quality scores, is frequently managed by the Trimmomatic software [24], and the software is also capable of removing adapter sequences. The TrimGalore package (https://github.com/FelixKrueger/TrimGalore, accessed on 15 February 2021) specializes in the removal of adapters and can also be configured to remove low-quality bases. The configurations tested here (described in Methods and contextualized in Figure 1) were intended to be typical for Trinity assemblies [25] rather than customized to these particular RNA-Seq experiments. The TrimGalore (TG) route removed far more reads on median in Sreedhar (6.8%) and Gupta (5.0%) than in Peláez (0.2%) and Wimberley (0.1%), suggesting that efforts to remove adapter sequences prior to the FASTQ export in the newer HiSeq 4000 instruments have largely been successful. See Appendix A for command-line details.

Trimmomatic (TM) was invoked as part of the call to Trinity. In two cases, Trimmomatic screened out a larger number of reads than did TrimGalore: The Peláez set was carved down by 1.93% and the Gupta set was reduced by 6.3%, while the other two were at essentially the same level. A further investigation of the reads retained by Trimmomatic revealed that some FASTQ files retained small numbers of adapter sequences. In the Wimberley set, we observed a considerably higher Trimmomatic read reduction in the Leaf2 and Leaf3 sets (removing an average of 31.3% of reads) than in the other four experiments (averaging 0.14% of reads).

Given the relatively small differences in read filtering, one might expect that Trinity de novo assemblies derived from these two routes would be quite similar. Instead, we observed that all the assemblies employing Trimmomatic contained more transcript sequences (of greater length) than the assemblies derived from TrimGalore FASTQs (see Table 2). The Wimberley data were particularly marked in their differences, with the TM assembly doubling the transcript sequence count of the TG assembly.

Because the Peláez data represented four different cultivars, each set of FASTQs were separately assembled. The assemblies ranged considerably in size, with Veracruz and Jalisco lagging Celaya and Cualac by a large margin. For many of the later analyses, we used Celaya as a sole representative from the Peláez set. The Celaya experiments accounted for 89,045,724 reads (28% of the total in Table 1 from Peláez). As the Gupta BioProject enumerated 39 FASTQs, we incorporated only the first of three files for each tissue to limit the RAM required for assembly. As a result, our analysis under-reports the sensitivity of their experiments.

These assemblies could become contaminated if species other than chia contributed RNA to the samples. MCSC Decontamination [26] compared clusters of transcript sequences to UniRef90 sequences to determine the likely taxa of origin and remove clusters deemed likely to have come from contaminating species. The number of transcript sequences remaining after this step was 74% of the original count in the TrimGalore assembly for Gupta, 85% for Peláez Celaya, 69% for Sreedhar, and 95% for Wimberley. As we were uncertain about the extent to which these reductions reflected spectral clustering or the removal of contaminant taxa, the original assemblies were passed to the next step rather than using those exported by MCSC Decontamination.

### 2.2. Determining Assembly Completeness

All other things being equal, one might naively assume that more reads imply a better sensitivity and, thus, more transcript sequences. This would suggest that Wimberley, with the most reads, should assemble to the most transcript sequences, with Sreedhar, Gupta, and finally Peláez Celaya ranking behind it. Neither TG nor TM assemblies follow that ordering, however. Sreedhar yields far fewer transcript sequences than any of the other three sets in both TG and TM assemblies (see Table 2). Gupta, Wimberley, and Peláez Celaya all vary in rankings for most transcript sequences in TG and TM assemblies, with a range from 108,164 to 301,858 transcript sequences. Helpfully, Trinity also organizes the transcript sequences into “genes” (roughly speaking, these are clusters of sequences that generate multiple transcript sequences). These cluster counts are far more consistent among assemblies, though Sreedhar gives evidence for roughly half as many “genes” as do the other data sets.

Benchmarking Universal Single Copy Orthologs (BUSCO) helps assess two key aspects of RNA-Seq derived transcript sequence sets: completeness and redundancy [27]. Each of the genes in a taxon-specific BUSCO database (eudicots in this case) is expected to appear in almost all member species, and each of the genes is expected to appear only once in the genome. Though Sreedhar consistently produced fewer transcript sequences and “genes” than the other data sets, its BUSCO completeness (summing complete single-copy and complete duplicated hits) rated it second among TM assemblies and third among TG assemblies (see Figure 2). “Concise” is more correct than “insensitive” in describing Sreedhar. BUSCO expects each gene to appear once in each assembly, and Sreedhar more frequently meets that goal than the other, more redundant assemblies.

Because the transcript sequence sets contain generally larger numbers of transcript sequences for Trimmomatic filtering than for TrimGalore filtering, it is possible that those added transcript sequences match a larger fraction of the BUSCO genes, boosting the completeness (see Appendix A). In all experiments except Gupta, the TM assembly achieves a higher BUSCO completeness than the TG, gaining as many as 136 more BUSCO genes in Sreedhar (out of a total of 2326 genes in the BUSCO eudicots set). Those gains, however, come with a cost; genes that should appear once in the set more frequently appear in multiple copies. In the case of Wimberley, choosing an assembly that contains twice as many transcript sequences seems an unreasonable cost to gain only 78 BUSCO genes. For subsequent stages of comparison, only the TrimGalore assemblies were retained.

The publication of the *Salvia splendens* genome sequence annotation [28] afforded the opportunity to recognize infrastructural RNAs such as rRNA and tRNA in these transcript sequence sets. The “GCA_004379255.1_SspV1” annotation contained 2208 sequences annotated as RNA-coding genes for *S. splendens*. After similar sequences in *S. splendens* and the four TrimGalore assemblies were clustered, reciprocal BLAST found relatively few transcript sequences to match at least one set of infrastructural RNAs from *S. splendens*: Six transcript sequences from Gupta, five from Peláez Celaya, six from Sreedhar, and eight from Wimberley. Infrastructural RNAs do not comprise a large segment of the assembled transcript sequence sets. The complete FNA assemblies are available in Appendix A.

### 2.3. Comparing the Assemblies

The Wimberley et al. paper was distinctive for including their assembled and annotated transcript sequence set in Appendix A. The published FASTA file contains 71,401 transcript sequences, with a median length of 1409 nucleotides (longer than any of the assemblies in Table 2). The much smaller number of transcript sequences reported by Wimberley et al. than assembled here likely resulted from their use of CD-HIT-EST clustering (reducing redundancy) [29] and requiring a match to a plant family ortholog. Using these published transcript sequences as an external reference can shed light on the assemblies created here; mapping reads to the assemblies constructed from them should match a large fraction of the sequences, while mapping them to an assembly that potentially represents a different cultivar should cut into the mapping rate.

Salmon performed quasi-alignments [30] for each TrimGalore-cleaned FASTQ to the assembly produced from that set of files and to the published assembly from Wimberley (see Appendix A). When mapping reads to the assembly derived from those reads, the Gupta set reads matched 85.6% of the time, while the other three experiments all matched more than 95%. One possible interpretation is that the plants sampled for Gupta were more genetically diverse than for the other three studies, with the genetic variants preventing the software from matching reads to the assembled transcript sequences. While the TrimGalore-cleaned FASTQs from Wimberley mapped very well to the assembly created here and to the assembly published by Wimberley (all values above 97%), the mapping percentages for the other three experiments to the published Wimberley sequences fell below 80%.

As these reads were ultimately produced from mRNA, genes with higher levels of transcription accounted for far more reads than did the others. Appendix A quantifies this phenomenon, determining the smallest number of transcript sequences required to account for 25%, 50%, 75%, and 100% of all mapped reads. For every pair of FASTQs among all four transcript sequence sets, fewer than 600 transcript sequences were necessary to account for the first quartile of mapped reads. The diverse tissues of Gupta illustrate the difference sample selection makes in read concentration. While ERR1855155 (Day 3 Cotyledon), ERR1855165 (Day 3 Shoots), and ERR1855210 (Seed) were extremely concentrated in just a few genes, ERR1855129 (Day 158 Raceme Top Half) and ERR1855202 (Day 69 Node) required an order of magnitude more genes to account for the first 25% of mapped reads.

Discerning the sequence overlap among assembled transcript sequence sets requires that we allow for sequence variants and truncations; two sequences from different assemblies may be homologous without being identical. ProteinOrtho software [31] determined which transcript sequences were the best match among the four best TrimGalore assemblies (using only the Celaya set from Peláez) and the published Wimberley transcript sequence set. ProteinOrtho clusters together sets of related sequences within an assembly before seeking orthologous sequences in another assembly.

ProteinOrtho reduced the numbers of transcript sequences in the four TrimGalore assemblies to 68–80% as many sequence clusters before comparison between assemblies. Of the four assemblies, the sequence clusters from Sreedhar were more frequently matched to orthologs in the other three (80%), while 57–63% of the clusters from the other three transcript sequence sets matched to one from at least one other assembly. The intersections of orthologs for the four assemblies plus the transcript sequences published by Wimberley et al. were visualized in an UpSet plot [32] (Figure 3). Of the 31 possible intersections among transcript sequence sets, only the most frequent twelve are visualized here.

It would be tempting to assume that a big chia assembly contains almost all sequences of a smaller chia assembly. To the contrary, the TrimGalore Sreedhar transcript sequence set, though smallest, contains 7157 sequences that appear in none of the other assemblies (eighth bar in Figure 3). Although all these sequence databases share a common core of 11,339 orthologs (sixth bar), each assembly contributes a considerable number of unique sequences. One might reasonably have expected that the intersection between the published Wimberley transcript sequence set and the one assembled here from the same data would have been the most common intersection between two databases (seventh bar), but sequences appearing in both the TrimGalore Wimberley assembly and the TrimGalore Gupta assembly were even more frequent (fifth bar). ProteinOrtho employed the BLASTN algorithm to compare these nucleotide sequences; a comparison at the protein level could appear quite different as many genetic variants (such as the “wobble” position in a codon) do not alter the protein sequences, and homology scoring differs as one shifts from nucleotides to amino acids.

### 2.4. Annotation via InterPro and Taxonomy Relationships

To produce a putative transcript sequence from de novo assembly is a good start, but determining its putative molecular function is necessary for the sequence database to be useful. We employed two avenues for functional determination: Seeking orthologs in the nearest taxonomic neighbors and checking for sequence motif matches in InterProScan. These “hits” could then be used to assemble an initial description line to accompany protein sequences in the FASTA file.

Detecting orthologs among taxonomic relatives evaluates the degree of detectable orthology among assembled sequences. At the time of writing, the order *Lamiales* contained nine species at NCBI with fully annotated genome sequences (featuring both genome-derived mRNA and protein sequence sets as described in Table 3). We six-frame translated the four “TG” transcript sequence sets, retaining every polypeptide of at least 100 amino acids. ProteinOrtho compared each transcript sequence or each protein sequence to those of the other species, seeking the clearest orthology relationships for each gene product. As the assembled transcript sequence sets were so much larger than the annotations to which they were being compared, the software was configured to exclude any gene product that lacked orthologs in any other species.

The orthology results for the four assembled transcript sequence sets, relying on BLASTN matching in ProteinOrtho, were unanimous; the most frequent orthology relationship discovered was between a transcript sequence in chia and a transcript sequence in *S. splendens* (data not shown). Given that *S. splendens* and *S. hispanica* were the only two species in the same genus, this is unsurprising. *S. indicum* and *H. impetiginosus* were universally found to have the next closest relationship. As others have frequently noted, nucleic acid sequence comparisons tend to highlight the nearest neighbors rather than distant evolutionary relationships.

Comparing among translated sequences via Diamond [33], however, detected the orthology across the *Lamiales* quite well. In all four of the assemblies, the most common or second-most common outcome was that protein sequences were associated with orthologs across the entire set of proteomes (even *A. thaliana*, which falls in a different order). A somewhat less common result was that orthologs were found among all species except for *G. aurea*, which features the most compact genome/shortest protein list of all the species in the comparison. In Figure 4, the ProteinOrtho/UpSetR plot for TGGupta demonstrates how many orthologs the following species contributed for gene products in *S. hispanica*: *S. splendens*: 2133, *O. europaea*: 1166, *G. aurea*: 1066, *S. asiatica*: 1025, and *P. japonicum*: 735. This ordering of species was universal among the four translated assemblies. After *P. japonicum*, the ordering of species varied among them. These ortholog pair counts illustrate the need to include many species in taxonomic databases used to attribute putative functions to novel sequences.

The orthology relationships with annotated genome sequences can yield information that ranges considerably in value. If a predicted chia protein sequence has an ortholog in *S. splendens*, for example, the *S. splendens* protein might be described only as a putative or hypothetical ORF. In this case, both *S. splendens* and *S. hispanica* annotations could be updated to reflect the conservation of this sequence between species. Discovering the orthology to a well-characterized protein, such as *A. thaliana* “cytochrome P450, family 714, subfamily A, polypeptide 2,” is far more informative. It is worth noting, though, that one may align multiple putative protein sequences from one species to a single ortholog in another, potentially highlighting gene duplication events or incorrectly fragmenting a transcript sequence to separate accessions due to incomplete sampling in RNA-Seq.

We sought to coordinate the information from InterProScan (motif matching) and from ProteinOrtho (orthologous sequences) into FASTA description lines that suggest potential annotations for the protein sequences. The Zorbit Analyzer software (https://github.com/dtabb73/Zorbit-Analyzer, accessed on 15 February 2021) can be configured to export only the translations that are orthologous to sequences from neighboring taxa or that match to sequence patterns recorded by InterPro, and it is able to track which sets of translations stem from an individual transcript sequence for reading frame determination.

In the case of the four translated assemblies constructed in this project, Zorbit Analyzer found that the great majority of sequences matching to orthologs in ProteinOrtho also matched to at least one InterPro signature (see Appendix A). An average of 46.6% of the translated sequences neither matched an ortholog nor matched an InterPro signature (this value is expected to be high because each transcript sequence was translated in six frames, though only the sequences that were longer than 100 amino acids were retained). While an average of 23.1% of the translations were matched to both InterPro and ProteinOrtho hits, only 1.0% matched via ProteinOrtho only. On the other hand, an average of 29.3% matched via InterPro but not by ProteinOrtho. InterProScan is much more likely to produce annotation information for a protein sequence than orthology hunting among nearby taxa.

### 2.5. Identification of LC-MS/MS Proteomes

Three distinct LC-MS/MS proteomes were employed to test the identification efficacy resulting from the different TrimGalore Trinity assemblies after the translation to amino acid sequences. Husselmann 2017 (1,000,863 MS/MS scans) was the Thermo Q-Exactive shotgun proteome that motivated this project, representing a trypsin digestion of total leaf protein from seedlings grown from either black or white chia seeds. Aguilar-Toalá 2019 (44,208 MS/MS scans) was a stress test for the databases, as the Alcalase and Flavourzyme digestion of seed proteins disallowed the use of trypsin specificity to filter database peptides. As these data were produced on a Thermo Q Exactive Plus mass spectrometer, the fragment ions were measured with high mass accuracy. Cooper 2021 (877,885 MS/MS scans) represented a broader investigation of chia tissues, including cotyledons, hypocotyls, roots, and seeds via simple RPLC and four-fraction RPLC on a Thermo Lumos, but it differed from the others in capturing low-resolution MS/MS scans and in using the closely related *Salvia columbariae* [34] rather than *Salvia hispanica*.

By incorporating the fractionation of three different tissues and benefiting from the ion trap scan rate of the Lumos instrument, Cooper 2021 generated the most sensitive proteome in this study, with nearly 33,000 distinct peptides identified in the best database search. The five-cohort Husselmann 2017 study incorporated almost twice as many LC-MS/MS experiments, but because it analyzed only one tissue and used no fractionation prior to LC-MS/MS, its most sensitive search identified fewer than 7200 distinct sequences.

The MSFragger database search algorithm [35] identified the Cooper 2021 and Husselmann 2017 sets of LC-MS/MS experiments five times; the first employed a translation of the published Wimberley transcript sequence set, and the next four each used a translation of a TrimGalore assembly. For a first look, we considered the total number of distinct peptides identified confidently from the aggregate of all RAW files in each of these proteomics data sets (see Appendix A; quality metrics appear in Appendix A). The sequence database providing the most sensitive search for Cooper 2021 identified 18% more peptides than the least sensitive, while the most sensitive search for Husselmann 2017 identified 24% more peptides than the least. In both cases, the best sequence database to support protein identification resulted from the six-frame translation of TGGupta, and in both cases, the sequence database producing the fewest peptides was translated from TGPeláez Celaya. It may not be coincidence that the TGGupta assembly spanned the largest number of tissues (13) while TGPeláez Celaya incorporated only seeds.

The use of different sequence databases resulted in different numbers of distinguishable proteins and a variation in empirical protein false discovery rates. For Husselmann 2017, the number of distinguishable proteins ranged from 1355 to 1634, with the maximum produced by the TGGupta search that yielded the most distinct peptide sequences (7145; see Appendix A), accounting for 170,492 PSMs (peptide-spectrum matches: An MS/MS with a confident peptide sequence assigned to it). The target/decoy method listed decoy proteins along with target proteins in the list, and an empirical protein False Discovery Rate (FDR) was estimated at 0.45%; all of the databases, in this case, produced an empirical protein FDR below 1%, with IDPicker 3.1 applying a conservative parsimony criterion and the two-distinct-peptide rule for protein inclusion [36]. Out of more than a million tandem mass spectra collected in Husselmann 2017, TGGupta made it possible to identify 17.03%, a respectable but not outstanding rate of identification.

The Cooper 2021 set also showed a considerable range in the number of distinguishable proteins, ranging from 4572 to 5443; again, TGGupta identified the largest number of proteins, supported by the largest number of distinct peptide sequences (32,923) and PSMs (241,993). The empirical protein FDR values were also somewhat higher than in Husselmann 2017, ranging from 1.09% to 2.19% (all generally within proteomics community expectations). The identification rate for Cooper 2021 rose to 27.57% of all spectra, which is the more impressive because the tandem mass spectra contained only low-resolution fragment ion data and represented sequence variants due to the *S. columbariae*/*S. hispanica* difference.

The protein sequences accounting for the most spectra in these two experiments will produce few surprises. In both sets, the sequence accounting for the largest fraction of all spectra was recognizable as a ribulose bisphosphate carboxylase large chain. The proteins ranked second by total PSMs in the two experiments differed, though their functions were inversely related: ATPase AAA core domain-containing protein (Husselmann 2017) and ATP synthase subunit beta, chloroplastic (Cooper 2021). Of these three, only the ATPase AAA core domain-containing protein was absent from the TrEMBL knowledge base.

One might reasonably expect that a transcript sequence set that incorporates the same tissue as a proteome would yield more identifications. The Cooper 2021 proteomes separately analyzed cotyledons, hypocotyls, and roots for five-day old seedlings plus ungerminated seeds. If we track which sequence database identified the most peptides for each LC-MS/MS experiment (see Appendix A), only root and seed proteomes yielded more peptides with databases other than TGGupta. Root tissue was not included among the 13 tissues of Gupta, but they represented half of the RNA-Seq experiments of Wimberley, which “won” in the proteomic identification for all five of the Cooper root LC-MS/MS experiments. These findings reinforce the idea that an RNA-Seq-derived transcript sequence set for a given tissue is better able to represent the proteins that may be identified by LC-MS/MS from that same tissue.

The Aguilar-Toalá 2019 experiment was not easily analyzed, because Alcalase/Flavourzyme cleavage is far less specific than trypsin. As “no enzyme” searches of large FASTA files in MSFragger require a prohibitively large amount of RAM, we opted to use the MS-GF+ algorithm [37] instead. The time required to search the <3kDa fraction (comprising 44,208 high-resolution tandem mass spectra) against these large protein sequence databases was at least five hours for each database (using a Xeon x5650 six-core processor).

This LC-MS/MS experiment, however, reveals the largest differences between best and worst identification sensitivity (see Appendix A). The TGWimberley database identified a low of 118 distinct peptide sequences in 144 PSMs, while the TGGupta database revealed 697 distinct peptides matching to 1068 different PSMs. The two RNA-Seq experiments containing reads from seed tissue (TGGupta and TGPeláez Celaya) identified more peptides than did the two RNA-Seq experiments that excluded seed tissue. When the peptide lists from these four experiments were overlapped in Venny (Oliveros, J.C.: https://bioinfogp.cnb.csic.es/tools/venny/index.html, accessed on 15 February 2021), however, the assembly from TGSreedhar contributed the most peptides distinctive to any single assembly (see Figure 5). As the overall number of identified proteins was much lower in Aguilar-Toalá 2019 (reaching a maximum value of 73 distinguishable proteins) than in the other two sets, including or omitting individual proteins from the protein sequence database had an exaggerated impact on identification sensitivity.

An examination of the Sreedhar-specific set revealed that four of the peptides from a single protein matched to a total of 18 spectra in Aguilar-Toalá 2019. NCBI BLASTP was able to align the protein sequence to UniProtKB, matching an uncharacterized *Salvia splendens* sequence and to *Sesamum indicum* 11S seed storage globulin/legumin B. In this case, choosing either of the seed-containing transcript sequence sets for identification would still have missed identifying a seed protein.

### 2.6. Evaluating Sequence Homology for Seed Storage Proteins and PUFA Synthesis

Because the Cooper 2021 proteome featured four different plant tissues, a spectral count table could illustrate proteins that were far more abundant in one tissue than in another. Seed storage proteins of chia are nutritionally interesting because these seeds are particularly high in protein content [38]. A very simple test sought proteins that matched more than 100 spectra in the seed proteome but less than 100 spectra in the three other tissues (this threshold is entirely arbitrary but helped focus attention on clear differences). The sequences of eleven proteins that met this “seed-specific” criterion from the TGGupta database have been included in a separate FASTA file in the Appendix A. All eleven proteins were matched by the BLASTP algorithm to the SwissProt database. Nine of the eleven can be putatively classified as seed storage proteins.

Given the interest in the chia synthesis of PUFA, it is unsurprising that Xue et al. sought to clone the FAD2 genes from this species [39]. We sought evidence of the expression for fatty acid desaturases from our assembled transcript sequence sets and identified proteomes. We identified the following integrated signatures in InterPro corresponding to this class of enzymes: IPR001522, IPR005067, IPR005803, IPR005804, IPR012171, and IPR021863. Of these, IPR001522 produced only two matches in any translated assembly, while the others ranged from 29 to 261 different transcript sequence matches (see Appendix A). These matches were frequently redundant, with multiple transcript sequences from the same Trinity “gene” cluster matching the same signature.

The two sequences published by Xue et al. matched both IPR005804 and IPR021863, leading us to emphasize those two signatures. IPR005804 was a more sensitive detector, with only 56 of its 146 transcript sequence matches also hitting IPR021863. IPR021863 matched to only three transcript sequences across the four assemblies that were not also matched by IPR005804. Ten ortholog sets were found to be present in all four assemblies while producing translations matching IPR005804. Clustal Omega produced multiple sequence alignments for each of these ten sets of orthologs (see Appendix A supporting these images).

Each of the IPR05804 alignments demonstrated a high degree of agreement among the translated sequences from each assembly, though an assembled transcript sequence would sometimes lack the N-terminus or C-terminus of the polypeptide. BLASTP was able to match to a sequence from either *Salvia splendens* or *Salvia hispanica* itself, in each case indicating a desaturase enzyme, generally differentiated by the specific substrate. The proteomics experiment from Cooper 2021 was used to see whether any of these fatty acid desaturases were detected. A variety of sequences from Sreedhar DN8237 were detected (36 spectra in a total of nine peptides drawn from three isoforms), joining proteomics and transcriptomic evidence for the presence of a sphingolipid delta(4)-desaturase (DES1-like). Similarly, the Gupta DN205 sequence cluster was detected in six spectra for three peptides that supported the detection of an omega-6 fatty acid desaturase (delta-12 desaturase), and eight spectra for three peptides supported the detection of a stearoyl-CoA desaturase (delta-9 desaturase). By combining RNA-Seq and proteomics evidence, the certainty of identification is greater than if either of these technologies were applied separately.

## 3. Discussion

Having produced two different assemblies for each of four RNA-Seq experiments in *Salvia hispanica*, we were surprised at the variation in the numbers of transcript sequences among these products (Table 2). One interpretation is that adapter sequences can confound the transcript sequence set assembly as repeats confound the genome sequence assembly; the adapter sequences may cause the assembler to overlap sequence reads that belong to completely different transcript sequences. Naturally, one might expect that a transcript sequence set assembled from a larger number of reads of greater length would contain more potential transcript sequences from each gene due to sensitivity differences. Given the diversity in the numbers of tissues sequenced for each experiment, we might have anticipated a greater difference in the comprehensiveness (estimated through BUSCO) of these assemblies than observed.

Once quality RNA-Seq data have been produced in a species, proteomics becomes possible. Understanding gene expression should not be limited to RNA-Seq. Measuring the proteome remains the only route to understanding the role of post-translational modifications. As protein turnover operates by different mechanisms than transcript turnover, quantitative proteomics is quite necessary to understanding which biological activities are at play in a system.

### 3.1. Is There Any Point in Working with RNA-Seq from Older Sequencer Models, Given How Quickly This Technology Evolves?

The three DNA sequencers employed in the above experiments were the Illumina Genome Analyzer IIx (an updated variant of an instrument first released by Solexa in 2006), capable of producing more than 2 gigabases of sequence per day; the Illumina HiSeq 2500, capable of more than 50 gigabases per day; and the Illumina HiSeq 4000, capable of more than 400 gigabases per day (based on manufacturer-provided specification sheets). As should be apparent from the BUSCO completeness presented in Figure 2 and in Appendix A, the Sreedhar RNA-Seq experiment achieved an admirable completeness despite using the oldest of the three sequencer models. A new sequencer that is multiplexed to analyze many barcoded samples at once may not outperform an older sequencer that can give more “attention” to a particular sample, especially if sample queues are less pressing on the older instrument.

### 3.2. If RNA-Seq Technology Is More Comprehensive Than Proteomics, Are All RNA-Seq Assemblies Good Enough for Protein Identification?

While LC-MS/MS gains considerable sensitivity with each passing year, the number of peptides detected in an experiment is dwarfed by the number of peptides available for sampling in a whole-cell lysate. The ability to quantify 10,000 proteins across tissues is still quite demanding in instrument time due to the need to fractionate and then subject each fraction to LC-MS/MS. In our Husselmann 2017 and Cooper 2021 proteomes, the potential gains for the best database versus the worst were 24% and 18% in aggregate distinct peptides. Those values represent the case where a very large number of proteins were present (1634 for Husselmann 2017 and 5443 for Cooper 2021). The seed proteome of Aguilar-Toalá 2019, however, concentrated all of its peptides in just 73 proteins. Including or failing to include a particular protein sequence could have an outsized impact on identification because the seed peptides were concentrated in a small number of protein sequences.

### 3.3. Is the Assembly of RNA-Seq Data within Reach for Most Proteomics Laboratories?

For newcomers to RNA-Seq to assemble their first sets of transcript sequences, they may not require many lines of script, but many challenges line that path. Our path to these assemblies started with a taxonomy search of the NCBI website followed by an investigation of the “Bio Projects” that had been posted for our species of interest. Our laboratory already had experience with Linux and had access to a computer with 96 GB of RAM, but proteomics laboratories frequently run their bioinformatics workflows on Microsoft Windows computers, and computers with more than 16 GB of RAM may be uncommon for proteomics facilities at present; without high-memory Linux workstations, Trinity is not an option. The collaboration between genomics researchers and proteomics researchers is likely to be the route to success for many teams.

Conversely, laboratories that are already fluent in RNA-Seq can add proteomics analyses to their repertoires by making use of the core facilities now available at many universities. The difference for nonmodel organisms will be that the laboratory needs to provide a protein sequence database in addition to the protein samples for analysis. At present, core facilities frequently rely upon the reference proteomes available through UniProt, which may under-represent plant proteins.

### 3.4. Is a Six-Frame Translation an Appropriate Search Space for Proteomics?

A single transcript sequence in an RNA-Seq assembly becomes six translations by considering each of the strands as coding and by having an uncertainty about which reading frame is correct for that strand. A given reading frame may have multiple long ORFs within it, leading to more ambiguity in the translations proposed from each transcript sequence. In species other than viruses, though, one generally expects that a given transcript is translated to produce just one polypeptide. Using a six-frame translation as input for proteomics substantially bloats the pool of potential peptides for comparison to tandem mass spectra. Most proteomics tools, at present, do not evaluate which of the six reading frames for a given transcript sequence is the dominant one, for example, by excluding identifications from the other five reading frames. Some statistical models for estimating false discovery rates explicitly assume that the number of decoy sequences in the protein database is equal to the number of target (potentially real) sequences; if one then takes the perspective that reading frames that are not the biologically significant one are also decoy sequences, the FDR computations will require significant adjustment.

## 4. Materials and Methods

### 4.1. RNA-Seq Source Data

**Sreedhar 2015**: A team with Malathi Srinivasan from the Central Food Technological Research Institute in Mysore, India evaluated five different stages of seed development in RNA-Seq using an Illumina Genome Analyzer IIx [20] (San Diego, CA, USA). Their Trinity-based de novo assembly yielded 76,014 transcript sequences; a FASTA file representing these sequences was not, however, included in their publication. The five sets of paired-end reads were released publicly as BioProject PRJNA196477 at NCBI.

**Peláez 2019**: A team with Angélica Cibrián-Jaramillo from UGA-Langebio Cinvestav in Guanajuato, Mexico evaluated genetic diversity and expression differences among the RNA-Seq experiments of eight different cultivated and wild chia seeds using an Illumina HiSeq 4000 [21]. Their Trinity-based de novo assembly spanned 69,873 transcript sequences; a FASTA file representing these sequences was not included in their publication. A total of sixteen sets of paired-end reads were released publicly as BioProject PRJNA448759 at NCBI. Our re-analysis included only the samples for which multiple sequencing experiments were performed: Spotted Celaya (3×), Cualac Celaya (2×), Spotted Jalisco (2×), and Spotted Veracruz (3×).

**Wimberley 2020**: A team with Hagop S. Atamian from Chapman University in Orange, California sequenced root and leaf cDNA using an Illumina HiSeq 4000 [22]. Their Trinity-based de novo assembly produced 103,367 transcript sequences; unlike the other studies, the team published a FASTA file with the paper documenting their effort. The leaf triplicates and root triplicates were each sequenced as paired-end reads, released publicly as BioProject PRJNA597830 at NCBI.

**Gupta 2020**: A team with Pankaj Jaiswal from Oregon State University in Corvallis, Oregon sequenced three replicates of thirteen tissues from different developmental stages of chia using an Illumina HiSeq 2500 [23]. Their Velvet/Oases assembly included 82,663 transcript sequences; their work, currently available as a pre-print, does not include the FASTA file enumerating the assembled sequences. All 39 paired-end experiments were released publicly as BioProject PRJEB19614. Our re-analysis included only the first of three paired-end replicates for each tissue.

All data sets were acquired through the use of the NCBI SRA Toolkit version 2.10.8 via the “prefetch” command (https://www.ncbi.nlm.nih.gov/books/NBK158900/, accessed on 15 February 2021). The “fastq-dump” command then exported the FASTQ files with these parameters: “--defline-seq ‘@$sn[_$rn]/$ri’ --defline-qual ‘+’ --split-files” to generate Trinity-compatible description lines and blank the quality description lines. The Sreedhar set required a reformatting of the description lines for Trinity via this operation: “awk ‘{{print (NR%4 == 1) ? “@1_” ++i “/1”: $0}}’”. FASTQC version 0.11.9 (https://www.bioinformatics.babraham.ac.uk/projects/fastqc/, accessed on 15 February 2021) generated a quality report for each FASTQ file produced by the four RNA-Seq experiments.

### 4.2. Trimming and Assembly

The de novo assembler used in this study was Trinity, version 2.11.0 [25]. Each assembly was produced using two different “trimmers.” The first employed Trimmomatic version 0.39 [24] as part of the command-line invocation of Trinity and excluded the use of Salmon. The second employed TrimGalore! version 0.6.6 (https://github.com/FelixKrueger/TrimGalore, accessed on 15 February 2021)/cutadapt version 2.8 [40] prior to Trinity, keeping only the reads for which the paired end was retained when adapter sequences and low-quality basecalls had been stripped away; this second pipeline employed Samtools version 1.10 [41], Salmon version 1.3.0 [30], Bowtie2 version 2.4.2 [42], and Jellyfish version 2.3.0 [43]. TrimGalore! was launched with this command line: “trim_galore --cores 6 --length 25 --paired --output_dir galore -q 5 -e 0.1 left.fastq right.fastq”, while Trimmomatic was run in-line with this addition to the Trinity command line: “--trimmomatic --quality_trimming_params ILLUMINACLIP:adapters/TruSeq2-PE.fa:2:30:10 LEADING:5 TRAILING:5 SLIDINGWINDOW:4:5 MINLEN:25”.

For Sreedhar, Trinity produced a single assembly spanning all five samples. For Peláez, Trinity produced separate assemblies for Celaya, Cualac, Jalisco, and Veracruz seeds to avoid the problems of assembly in the presence of extensive sequence variation [44]. For Wimberley, Trinity produced a single assembly spanning the triplicates of roots and triplicates of leaf tissues. For Gupta, Trinity produced a single assembly spanning the first replicate for all thirteen tissues. Only the Sreedhar assembly was small enough to be assembled by a Linux workstation with 16 GB of RAM; the others required servers with at least twice that capacity.

CD-HIT clustering is frequently applied to de novo assemblies to reduce the redundancy of assembled sequences [29]. It can, however, mask short isoforms that may be produced for a gene in favor of longer transcript sequences. It was omitted from this workflow.

MCSC Decontamination [26], downloaded on 12 March 2021, was applied to evaluate the extent of species other than chia in RNA-Seq assemblies. The UniRef90 providing sequences for comparison was updated on 30 June 2020. MCSC was configured for clustering level 6, with a target taxon of “Chlorobionta.”

BUSCO version 4.1.4 [27], supported by Augustus 3.3.3 [45], evaluated the completeness of each assembly. The software employed the “eudicots” lineage dataset (10 September 2020 version) and operated in “transcriptome” mode. NCBI BLAST version 2.11.1+, specifically tblastn, powered the homology search.

### 4.3. Assembly Mapping

Trinity includes a Perl script titled “align_and_estimate_abundance” to determine the number of reads mapped to genes and transcript sequences of a given assembly. This script, in turn, launched Salmon v1.3.0 to prepare the reference database indexes and then estimate the abundance of genes and transcript sequences. The Salmon “quantmerge” software then combined the salmon mappings from the different reads that comprised each data set into a single table. In each TrimGalore Trinity assembly, Salmon was configured to operate in “trinity_mode” to retain the information of which transcript sequences corresponded to each gene (except for the published Wimberley transcript sequence set).

### 4.4. Transcript Sequence Overlap

The same transcript sequence in multiple assemblies might differ in which exons were detected or may be homologous rather than identical if a different cultivar of chia were grown. ProteinOrtho version 6.0.24 [31] detected the best matches among transcript sequence sets, using the NCBI BLAST version 2.11.1+ [46] engine for scoring matches (specified by the “-p=blastn” option). The -singles option ensured ProteinOrtho included unmatched sequences in its report. The software generated its table of matching sequences in both tab-separated values and interactive HTML reports.

A script in R statistical environment version 3.6.2 [47] read the TSV report from ProteinOrtho and determined which transcript sequences (rows) contained an accession for each transcript sequence set (columns), with an asterisk marking a transcript sequence that was not detected in a transcript sequence set. The scripts made use of the UpSetR library [32] to visualize the transcript sequences overlapping among transcript sequence sets. The R script is included as Appendix A.

### 4.5. Protein Sequence Annotation

Tools for functional annotation frequently emphasize protein sequences rather than transcript sequences. For these and later analyses, we limited our examination to the more compact TrimGalore assemblies and chose the Peláez Celaya assembly to represent that project rather than considering the full set.

EMBOSS version 6.6.0.0 [48] offers the transeq and checktrans tools for this purpose. Transeq accepts a transcript sequence FASTA and outputs a protein sequence FASTA in six frames via the “-frame 6” option. Checktrans screens these protein FASTA databases to report only the ORFs of at least 100 amino acids via the “-orfml 100” option. Setting the length threshold at 100 greatly reduces the number of spurious translations included in output, but it also has the effect of screening out short protein sequences. For context, the NCBI *S. splendens* protein database uses a minimum length of 50 amino acids; 1950 (3.6%) of the proteins are 100 amino acids or fewer, and 6491 (12.1%) are 150 amino acids or fewer. As checktrans has the side effect of removing description information from FASTA accession lines, we also created a short Python script capable of retaining this information (https://github.com/dtabb73/SixFrame-Translate, accessed on 15 February 2021).

Attributing a function to a novel transcript sequence or its translated product generally depends upon establishing homology to a better annotated sequence. As above, ProteinOrtho was employed to find orthologous proteins for each translated assembly. Where these examinations took place among amino acids, ProteinOrtho was able to use the high-speed Diamond version 0.9.30.131 homology aligner. For comparing transcript sequence sets, ProteinOrtho employed NCBI BLAST+ version 2.11.1+. As in the assembly comparison above, the -singles option was employed so that unmatched proteins were retained in the ProteinOrtho output. The complete protein databases at NCBI from the order *Lamiales* that were included in this comparison appear in Table 3. Due to the maturity of its annotation, the more distantly related *Arabidopsis thaliana* was included as a model. Transcript sequence sets for each of these species were downloaded, where provided, and where necessary, the transcript sequence sets were recreated from the complete genome sequences by gffread version 0.12.3 [49] with this command line: gffread -w transcripts.fna -g genome.fna transcripts.gtf. The transcript sequence set for *S. asiatica* was excluded because multiple transcript sequences bore identical accessions.

The UpSetR script in Appendix A includes the code used to produce these intersection plots; a key difference is that all protein/transcript sequences that were unique to only one species were eliminated from the table prior to visualization (the assemblies contained far more sequences than the FASTAs derived from genome sequence annotations).

InterProScan v5.47–82.0 [50] screened the translated assembly sequences for sequence motifs and protein family relationships. SignalP v5 [51] and TMHMM v2 [52] supported this motif search with tools to recognize signal peptides and transmembrane domains. In order to accelerate this matching, the amino acid FASTA databases were split into sub-databases of no more than 10,000 sequences by FASTA File Splitter version 1.0.5381 (available from Pacific Northwest National Laboratory at https://omics.pnl.gov/software/fasta-file-splitter, accessed on 15 February 2021). InterProScan exported its reports to GFF3, JSON, XML, and TSV formats.

In order to analyze the combined outputs of InterProScan and ProteinOrtho, a Python script using a Pandas database (https://pandas.pydata.org/, accessed on 15 February 2021) was developed. The Zorbit Analyzer (https://github.com/dtabb73/Zorbit-Analyzer, accessed on 15 February 2021) integrates information from the translated Trinity assembly in FASTA format, the ProteinOrtho TSV file, and the InterProScan GFF3 file. Accessions in the protein sequence database can be matched to the accessions in the ProteinOrtho table and in the InterProScan GFF3. The Pandas library creates a database joined by these three sources of information, using the accession number as the key. It can then write a new description line for each protein accession that yields orthologs in ProteinOrtho or domain matches in InterProScan, outputting the protein FASTA database at the script conclusion. The code employs the numpy and pandas libraries and was developed in Python 3.9.0.

### 4.6. Proteomics Source Data

**Husselmann 2017**: Following Williams’ gel and TOF/TOF investigation of chia salinity stress (http://hdl.handle.net/11394/5650, accessed on 15 February 2021), Husselmann and Klein designed an LC-MS/MS experiment for five replicates in each of five cohorts: Caffeic acid stress, salt stress, both stresses, and controls for black chia seeds, along with controls for white chia seeds. Extracted proteins were analyzed at the Centre for Proteome and Genome Research. L. Bell denatured proteins, reduced disulfides with TCEP and alkylated cysteines with MMTS, and digested proteins with trypsin in a HILIC magnetic bead workflow. The RPLC gradients yielded an average of 34,512 tandem mass spectra per LC-MS/MS experiment on the Thermo Q Exactive (San Jose, CA, USA). Quadruplicates of samples pooling all cohorts were produced for chromatographic peak normalization. Detailed methods can be found in the Appendix A. The RAW files and representative identifications are available from ProteomeXchange as MSV000086861.

**Aguilar-Toalá 2019**: Investigators with Andrea Liceaga at the Department of Food Science at Purdue University examined the proteomes of chia seeds for their anti-microbial properties and enzyme inhibition [18,19]. They followed the activity for different size exclusion chromatography fractions of peptides generated through Alcalase and Flavourzyme digestion. Tandem mass spectra were generated on a Q Exactive Plus by Emma Doud in the Proteomics Core at the Indiana University School of Medicine. The LC-MS/MS experiment representing proteins below 3 kDa was emphasized because it represented the biological activity they sought; the instrument produced 44,208 tandem mass spectra at a rate peaking near 11 Hz. When sequence availability for chia proved to be problematic, Doud made use of PEAKS Studio (Waterloo, ON, Canada) to infer sequences directly from the fragment ions. The RAW files and representative identifications are available from ProteomeXchange as PXD024163.

**Cooper 2021**: Cooper sought to increase the sensitivity of peptide detection and broaden the range of tissues covered through his analysis of chia proteomes. As both *S. hispanica* and *S. columbariae* are both colloquially called “chia”, a miscommunication led to the preparation of *S. columbariae* proteomes rather than *S. hispanica* proteomes. Targeted tissues were ungerminated seeds and the roots, hypocotyls, and cotyledons of five-day-old seedlings. As in Husselmann 2017, TCEP was employed for reducing disulfides, but iodoacetamide alkylated the cysteines. Mass spectrometry took place at the Mass Spectrometry and Proteomics Facility at the Johns Hopkins School of Medicine. The Thermo Orbitrap Fusion Lumos Tribrid mass spectrometer yielded an average of 54,868 tandem mass spectra per LC-MS/MS experiment, producing up to twenty MS/MS per second at its peak. Detailed methods can be found in the Appendix A. The RAW files and representative identifications are available from ProteomeXchange as PXD024181.

### 4.7. Proteomic Identification

Each proteomic dataset was analyzed via quality metrics and via database search-based identification. As an initial step, Thermo RAW files were converted to mzML format in ProteoWizard msConvert 3.0 [53], using --zlib to compress peak intensities and *m*/*z* values and --filter “peakPicking true 1-” to centroid all peaks. QuaMeter IDFree generated quality metrics from these mzML files [54]; the tables are available in Appendix A.

In each case, the amino acid FASTA databases were supplemented with contaminants and were doubled to contain each sequence in both normal and reversed orientation (the latter serving as decoys for FDR estimation). This operation was automated by the Philosopher database [55] commands. The MSFragger database search engine [35] was operated in “narrow” search, using a precursor mass accuracy of 20 ppm, and semi-trypsin specificity (requiring each peptide to either begin or end at a trypsin cutting site). The software was configured to consider Met residues in normal or oxidized (+16 Da) form, and protein N-termini acetylation was allowed (+42 Da). For Husselmann 2017, Cys residues were all expected to be +46 Da, reflecting the use of MMTS, while Cooper 2021 Cys residues were expected to be +57 Da, reflecting the use of iodoacetamide. The two searches differed in that the fragment mass tolerance for Husselmann 2017 was set to 20 ppm while Cooper 2021 required a much wider 0.5 Da tolerance for fragments.

The Aguilar-Toalá 2019 set used cleavage enzymes with a much broader specificity than trypsin. MS-GF+ [37] was employed to produce this search rather than MSFragger because MS-GF+ requires far less RAM for this type of operation. The configuration was similar to the search for Husselmann 2017, though MS-GF+ uses “Instrument ID” set to “Q Exactive” rather than specifying the fragment tolerance directly. The “Enzyme ID” should ideally have been set to Alcalase and Flavourzyme, but this option has not been defined for MS-GF+; instead, it was set to “Chymotrypsin,” and the “Number of Tolerable Termini” was set “NTT = 0,” implying that neither end of a given peptide needed to meet the chymotrypsin specificity requirement for the peptide to be matched to spectra. Cys was left at its standard mass rather than adding a mass to reflect an alkylation step.

Database search results were imported from pepXML or mzIDentML format into the IDPicker 3.1 protein assembler [36]. Default filtering options held the PSM FDR to no greater than 2% (based on target-decoy analysis) for each LC-MS/MS experiment. Two distinct peptides were required for each protein to be included, and parsimony was employed.

## Figures and Tables

**Figure 1 plants-10-00765-f001:**
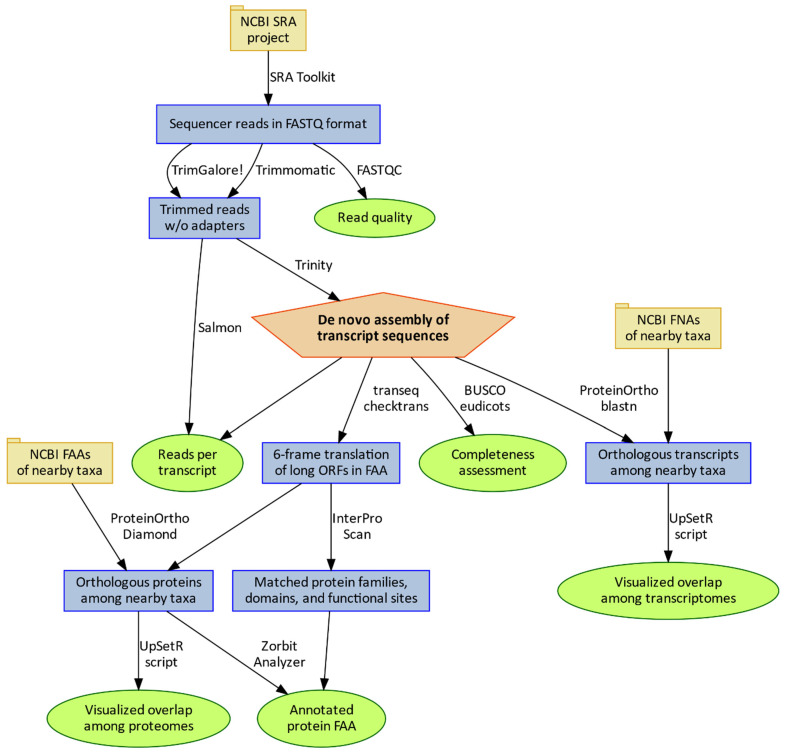
Once sequencing reads have been assembled to a set of transcript sequences (represented by the orange pentagon), a great variety of analyses become possible, with each information product represented by a green ellipse.

**Figure 2 plants-10-00765-f002:**
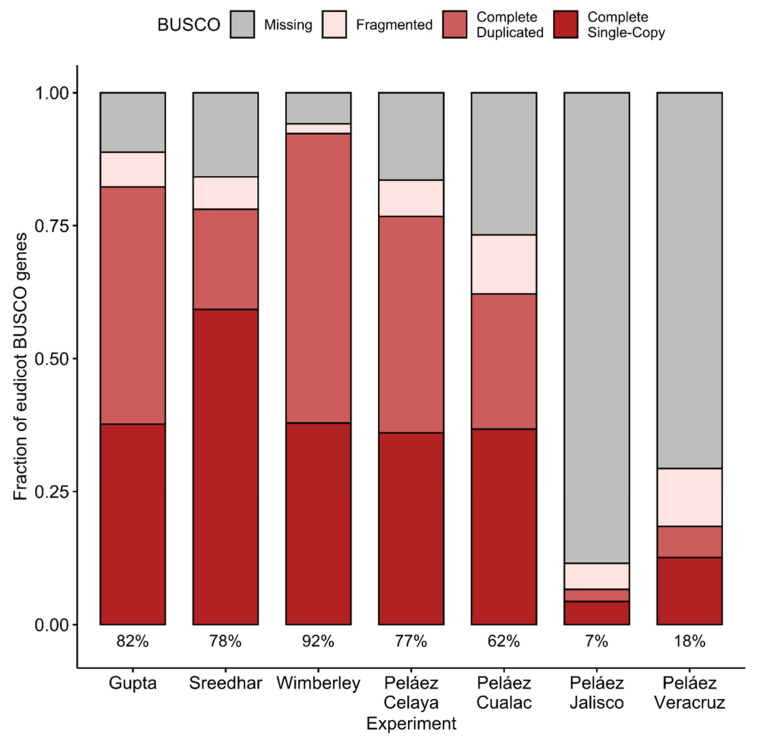
BUSCO assesses whether or not single-copy orthologs are represented in an assembled transcript sequence set (“completeness”) and whether or not they are present in multiple copies (“redundancy”). The first five bars illustrate a high completeness (red plus grey: Percentages shown below bar) but also considerable redundancy (grey bar) within the TrimGalore assemblies; the last four bars illustrate the heterogeneity among the four cultivars in the Peláez set.

**Figure 3 plants-10-00765-f003:**
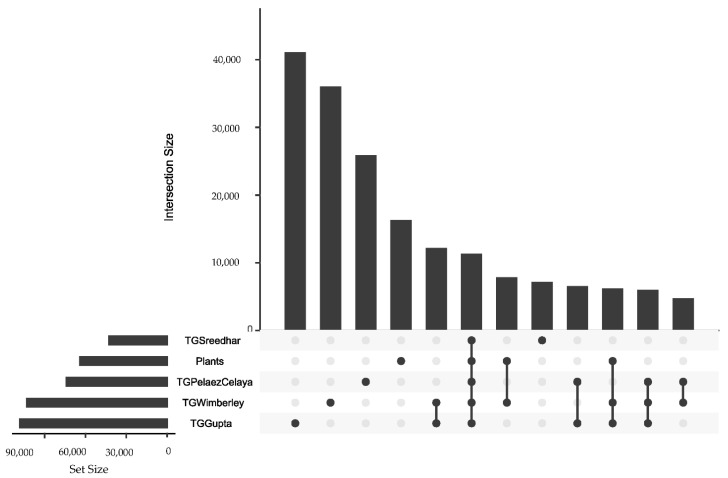
UpSet plot visualizing orthology among assemblies. Each transcript sequence set is internally clustered; its size after clustering is represented by the bar size in the lower left. The sizes of intersections (sets of sequences found in multiple assemblies) are shown by bars in the main plot, sorted with the most common intersections first. The beads connected by lines below report which assemblies contain each set of homologous sequences. “Plants” represents the transcript sequence set published by Wimberley et al., while “TGWimberley” represents a new Trinity assembly of the Wimberley FASTQs, pruned by TrimGalore.

**Figure 4 plants-10-00765-f004:**
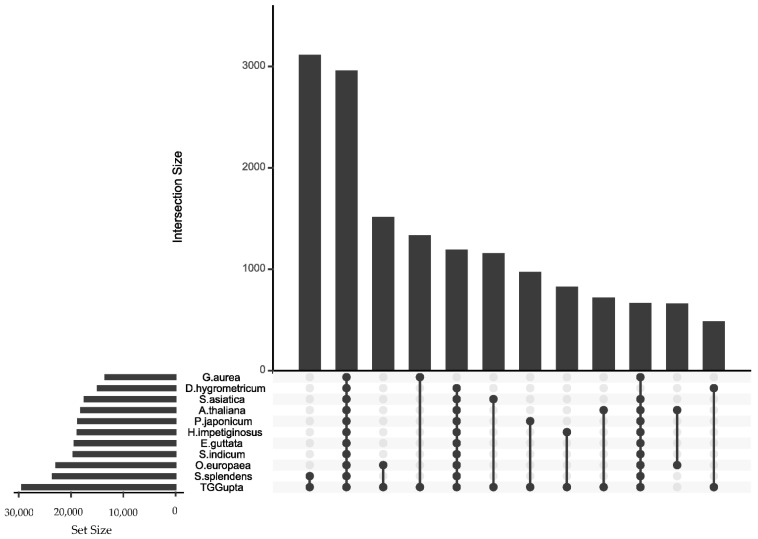
Like Figure 3, this UpSet plot shows the extent of the orthology between sequence databases, but in this case, the diagram visualizes the homology among protein sequences rather than transcript sequences, and the comparison ranged across *Lamiales* rather than between assembled *S. hispanica* transcript sequence sets. Clusters of protein sequences found in only one species were excluded. The TGGupta transcript sequence set, when translated to amino acids, showed a strong homology to proteins of other taxa within *Lamiales*. The number of orthologs found across the entire set of species was almost as large as the number of orthologs found uniquely with its genus-mate *Salvia splendens*.

**Figure 5 plants-10-00765-f005:**
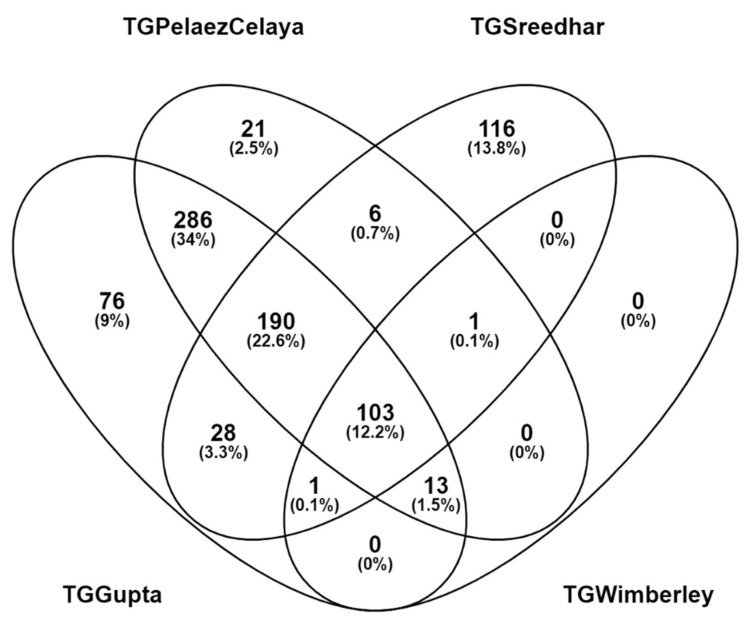
Different sets of peptides are identified from Aguilar-Toalá 2019 in each of the four translated assemblies. The two assemblies that incorporated seeds (TGGupta and TGPeláezCelaya) identified the most peptides, but the assembly that added the most peptides uniquely was TGSreedhar.

**Table 1 plants-10-00765-t001:** Trinity assembled each of four different chia RNA-Seq studies individually.

	Sreedhar	Peláez	Wimberley	Gupta
Accession	PRJNA196477	PRJNA448759	PRJNA597830	PRJEB19614
Expts (pairs)	5	10 *	6	13 *
Tissues	Developing Seeds	Seeds	Leaves and Roots	Thirteen Tissues
Sequencer	Genome Analyzer IIx	HiSeq4000	HiSeq4000	HiSeq2500
Total Reads	178,652,566	314,945,194	224,284,401	138,921,028
Read Length	72	99	99	99

* This analysis included only cultivated strain data from Peláez; similarly, it included only the first of three FASTQ pairs from each tissue for Gupta.

**Table 2 plants-10-00765-t002:** The RNA-Seq assemblies ranged from 27,543 to 301,858 transcript sequences, with the trimming tool playing a large role in the number and length of transcript sequences. “Genes” is reported by TrinityStats, representing the number of sequence clusters from which transcript sequences were drawn.

	TrimGalore!			Trimmomatic		
Assembly	“Genes”	Transcript Sequences	Median Length	“Genes”	Transcript Sequences	Median Length
Gupta	72,525	145,679	660	61,217	156,748	845
Sreedhar	34,590	53,127	857	30,649	63,468	1129
Wimberley	71,869	142,899	837	84,087	301,858	1388
Peláez Celaya	60,319	108,164	756	56,761	186,842	1319
Peláez Cualac	41,802	71,324	775	39,705	109,835	1162
Peláez Jalisco	20,106	27,543	357	20,908	30,909	377
Peláez Veracruz	36,749	50,681	427	38,666	59,163	469

**Table 3 plants-10-00765-t003:** The completed genome sequences within the order *Lamiales* range widely in transcript sequence set size, from 17,685 transcript sequences for the carnivorous *G. aurea* to 67,009 transcript sequences for the European olive. [1] *A. thaliana* falls within the order *Brassicales* rather than the order *Lamiales*. [2] The transcript sequence set for *S. asiatica* contains redundant accessions that greatly outnumber the proteins listed for this species.

Species	Transcript Sequences	Proteins
[1] *Arabidopsis thaliana*	53,827	48,265
*Dorcoceras hygrometricum*	47,778	47,778
*Erythranthe guttata*	34,410	31,861
*Genlisea aurea*	17,685	17,685
*Handroanthus impetiginosus*	30,271	30,271
*Olea europaea*	67,009	58,334
*Phtheirospermum japonicum*	30,299	30,330
*Salvia splendens*	55,562	53,354
*Sesamum indicum*	38,621	35,410
*Striga asiatica*	[2] 100,380	33,426

## Data Availability

Transcript sequence set reads are available from the following BioProjects at NCBI Sequence Read Archive: PRJNA196477, PRJNA448759, PRJNA597830, and PRJEB19614. Proteome LC-MS/MS raw data are available from MassIVE and ProteomeXchange at these accessions: MSV000086861, PXD024163, and PXD024181. The TGGupta amino acid sequence database accompanies these proteomic data. Appendix A available from the journal website includes the Appendix A mentioned above, plus the four Trinity assemblies in both FNA (nucleic acid FASTA) and FAA (amino acid FASTA), a small FAA database for seed-associated proteins, and small FAA databases for ten putative fatty acid desaturases.

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
