# Peer review of "Proteomic Identification and Meta-Analysis in Salvia hispanica RNA-Seq de novo Assemblies"

_plants, 2021, doi:10.3390/plants10040765_

Round 1
Reviewer 1 Report
It is good ideal to compare the values and adaptability of different software. But there some suggestions to the experiment:
- English is too hard to read,author should revise the language before it was published.
- The authors try to use different software to analyze different proteomics and metabolomic databases and to evaluate the reliability and value of these software. But, the whole study was descriptive introduction. Authors can design some experiment to verify its reliability, e.g. by q-PCR and sequencing. Also, the authors can compare their own answers with others' to prove the reliability of their conclusions.
- The analysis of proteomics and metabolomics have not been well combined. It is an individual. Maybe, just like two experiments. Maybe, the authors can analyze the database of same treatment experiment, so the two parts of the paper can be well combined.
- Figure 5 When analyzing the same metabolic group with different software(Fig5), the conclusions obtained by the software are very different. How to explain it, and hot to carry out a better analysis and provide solutions?
- Can the author design one experiment to verify its reliability of proteomics and metabolomics?
Author Response
- Thank you for drawing our attention to the language we used. In reading the entire manuscript aloud, we found several places where better language was needed. We hope that the revisions make all sections more accessible.
- Since this project is largely bioinformatic in nature, we believe that the advances we are presenting in data analysis are the "center pole" of the manuscript. That said, this manuscript is introducing two new "bench" proteome experiments (Husselmann 2017 and Cooper 2021) that have not previously been published. We have sought in this revision to emphasize valuable information for plant biology rather than limiting ourselves to methods development.
- We have sought in the new section 2.6 to incorporate both transcriptome and proteome data in our work with seed proteins and fatty acid desaturases. Showing that these technologies complement each other is certainly part of what we seek to establish with this manuscript.
- It is certainly the case that the presence or absence of particular proteins in a database can cause large variances in identification performance when the sample includes fewer than one hundred proteins. We hope that we have explained this phenomenon better in the revised manuscript.
- While ten days for major edits did not leave sufficient room for added bench biochemistry, we hope that the new analyses we have added and concrete uses of proteogenomics to assemble new information on seed proteins and fatty acid desaturases will be satisfying. We believe that the tables of identifications in Supplementary Tables 5 and 6 demonstrate the reliability of proteomics, and we were very encouraged about the reproducibility of transcript inference, as illustrated in Figure 3.
Reviewer 2 Report
Klein et al. have performed the Proteomic identification and meta-analysis in Salvia hispanica RNA-Seq de novo assemblies. The method section is written in detail and the code is provided as needed but it made the method section longer. The author can make the GitHub page and deposit their code there instead of the method/supplementary section. The author trying to expand the scope of transcriptomic data analysis to the level of proteome diversity, considering proteome as a route to understanding the post-translational modifications. The overall goal was good, but it lacks the biological information from the analysis. The author missed the opportunity to discuss the biological meaning of the analysis. I think that is the most important part for plant biologists. The authors have nicely used the publicly available transcriptome and proteome datasets. This is a good example of utilizing the flood of the genomic dataset generated these days.
I have divided my comments into major and minor comments.
Major comments:
- The number of transcripts per assembly is not anywhere close to each other. Authors need to give a comparison with the published assembly for the datasets they used. Further, the author needs to give a brief procedure on how they filter the initial assembly generated by Trinity. For example, whether minimum read support was considered to be a transcript. Contaminants were removed or not? It can be done with BLAST against different organisms other than plant samples like microbes, arthropods, humans, mice, etc., etc. The author did run the CD-HIT to remove the redundancy but did they consider the minimum length of ORFs as well. Is the transcript column only representing the protein-coding transcripts or non-coding as well in table 2? Maybe separating coding/non-coding will reduce the number of transcripts per assembly, if it’s not done yet. I suggest using some type of filtering if they haven’t used it already. If they have used present it clearly how/where it was used. The following paper has done the filtering and the author can follow similar steps as mentioned in the paper.
Fan L, Wang G, et al. (2018) Transcriptomic view of survival during early seedling growth of the extremophyte Haloxylon ammodendron. Plant Physiol Biochem 132: 475–489
- Figure 2. The information about the total number of ortholog groups/pairs was buried somewhere. Authors need to present clearly how many ortholog group/pair was identified from this study? Also, it will be better to go into the biological side of the analysis. What were those ortholog pairs related to? Authors need to make some supplementary figures to show the functional enrichment of those orthologs.
- Section 2.2 and 2.3 can be switched. It will be good to have the quality assessment of the assemblies before using those assemblies for analysis. This will further require switching figures 2 and 3.
- In figure 3, there is a huge difference in the complete single-copy transcripts between the assemblies/cultivars. The authors need to have an explanation of why there is such a difference existed? Is there variation between the cultivars in the ploidy for Salvia hispanica? Authors need to include the % of each category within the graph. For example, x% complete single copy, y% duplicated copy, etc., etc.
- The result section is very difficult to follow. Mention figure/table/supplementary figure/supplementary table as much as possible in the main text. I think that will help guide the reader.
- Is the accession/cultivar used for transcriptome and the proteome the same? If not, the author needs to clarify why they are comparing two different cultivars. I assume, at least cultivars need to match if they are doing the comparison between datasets if not the tissues.
- What is the main finding of the proteome data analysis? It is not mentioned clearly if and not discussed either.
- Author need to discuss their finding in detail. The discussion is like a Q/A right now. I suggest the author discuss the biological finding of this study. What is the significant message of the paper author trying to convey? I could not identify that part. How the finding is novel and connected to the nutritive value or polyunsaturated fatty acid production in Chia.
Minor comments:
- Citation style is not matching the format. Line 34 and all other places; nutritive value[1]. Need to change the format throughout the manuscript as nutritive value [1].
- Modify table 1, Column 2, what is the tissue type for Sreedhar? Is it a leaf during seed development? Make it clear. Column 3, seeds of cultivars not needed just use seed.
- Define set size in both the UpSet plot in Figures 2 and 4.
- What is the use of table 3? Why it is there, I could not follow that because similar data is presented in figure set size.
- The number of transcripts does not match with the set size in figure 4, what is this number?
- Line 271-273, what are these numbers? Should this number match somewhere in figure 4? It is not matching.
- Line 363-367 sentence not clear.
- Line 380 indentation missing.

Author Response
We would like to start by saying "thank you" to this reviewer. Having so many concrete points of advice helped establish a pathway for how we used the short time allowable for revision. As newcomers to the field of transcriptomics, we benefited tremendously from this guidance. We enumerated our "must have" list to included these analyses:
- We sought a tool that could recognize potential non-chia contaminants in our transcriptome. We identified MCSC Decontamination as a solid choice. The final paragraph of Section 2.1 showed that a substantial number of the transcripts assembled in our experiments could be screened out as contaminants (for example, we MCSC Decontamination retained only 69% of the transcripts in Sreedhar and 74% of the transcripts in Gupta). The extent to which this resulted from clustering of related transcripts rather than contaminant removal was somewhat opaque in the software's output, though.
- We prioritized recognizing RNA sequences that did not code for proteins. Given the availability of RNA sequences from Salvia splendens, we sought homologous sequences in the Trinity assemblies for Salvia hispanica. A surprisingly small number of sequences were matched, though, suggesting that the methods for emphasizing mRNA were working reasonably well for all four RNA-Seq experiments. A new paragraph at the conclusion of section 2.2 addresses these findings.
- The reviewer was correct that the number of reads supporting individual transcripts was quite unevenly distributed. We elected to show quartiles of our mapped reads, asking the smallest number of transcripts required to explain a given quarter of mapped reads. The second paragraph of section 2.3 now addresses this distribution, and Supplementary Table 3 provides the data behind it.
- The reviewer noted that we needed to spend more time with the overlaps among our transcriptomes to show reliability. We focused on this question: "What fraction of the transcripts in this assembled transcriptome are also observed in other transcriptomes?" We added material to the fourth paragraph of Section 2.3. to answer this question.
- The most significant addition to the revised manuscript is the all-new Section 2.6, supported by additional FASTAs in the Supplementary Information and a set of ten multiple-sequence alignments as Supplementary Figure S1. What did we learn about plant biology from this new arsenal of information? We learned that many fatty acid desaturases (including FAD2) were universally detected from the transcriptomics experiments. We have shown the polypeptide sequences representing each of the assemblies in highly-conserved alignments that match the same InterPro signatures as do the FAD2 sequences cloned by Yufei Xue et al. We also used proteomics expression data to enable us to recognize seed proteins, all of which have orthologs in other species. We believe that this section will be of great value to plant biologists in determining whether these transcriptome assemblies are of value.
The reviewer marked several other places in the manuscript that we should examine. Among other fixes, we have re-rendered the figure of BUSCO scores to make its information easier to absorb. We decided to move our six-frame translation code to GitHub rather than including its source code in Supplementary Materials. We did, however, retain the table showing genome information for nearby taxa because we felt it was necessary for readers to be able to see the range of genome sizes observed for neighbors to Salvia hispanica.
Again, we greatly appreciate the advice from reviewer 2.
Reviewer 3 Report
The manuscript is well written and therefore it can be accepted for publication.
Author Response
We appreciate the vote of confidence in our work. Thank you!
Round 2
Reviewer 1 Report
- The authors try to revise the manuscripts with better language, but I think they still need to be revised before it is published.
- The reliability has been verified by some analysis.
Author Response
I appreciate that Reviewer 1 was able to render feedback on our manuscript as quickly as he or she did. We made a great many changes to the manuscript language in response to the first round of reviews, so much so that the "tracked changes" in Microsoft Word touched almost every paragraph. We made considerable edits to the proteomics paragraphs because we found many of them used confusing wording.
At this point, I would request clarification of the language feedback. Since the author team is more familiar with proteomics than transcriptomics, are we using unusual language to describe our work with RNA-Seq? Are we writing with a bioinformatics emphasis rather than a plant biotechnology one? From an English language perspective, is our use of first-person plural rather than passive voice undesirable? Are there particular sections that are likely to confuse readers?
At this point, we are unsure which aspects of language and style to address.
Reviewer 2 Report
The author could have highlighted the revision in the manuscript.
- Y-axis in figure 2 is not clear, is it % of genes for each BUSCO category?
- The title for result section 2.6 “validating” is not a suitable word since it was not experimentally validated in this report. Change the word.
- Table 1, Column 3- just use seed.
- Is there any study author can mention showing the ploidy level variation between the cultivars for Salvia hispanica? This will help to justify the variation of single-copy and duplicated genes in their transcriptome assemblies.
- Define set size in both Figures 3 and 4. Is it a total number of protein-coding transcripts?
Author Response
We apologize for sending only the finalized version of the Word document and not the version with all the tracked changes. In preparing the new revision, we made some of the edits before remembering to turn tracking on. This is the detailed list of changes for this revision:
- We used more direct wording for the closing text of the Abstract.
- We wrote more directly about the bean proteome in the 4th paragraph of the Introduction.
- We altered "tissues" for Pelaez in Table 1.
- We changed from "disparate" to "orthogonal" in the first paragraph of 2.1.
- We dropped the blue and green coloration in Table 2.
- We replaced the image on Figure 2 with one that included more detail on the y-axis.
- Figures 3 and 4 were replaced with SVG versions, and context was added to both captions.
- The Section header for 2.6 was reworded.
- We slightly altered the wording employed in Section 4.6.
We have corrected the y-axis for Figure 2 in the new revision.
We agree with Reviewer Two that "validating" was the wrong word choice for the 2.6 header. We have opted for "Evaluating sequence homology for seed storage proteins and PUFA synthesis" for the new header on section 2.6.
We altered the Pelaez tissue to read "seeds" as suggested, rather than specifying we were using only the cultivars and excluding the wild varieties from Pelaez.
We agree that the topic of ploidy is one possible reason why some of these transcriptomes might offer more transcripts while other would offer fewer. When I read through a few papers on this topic in the Salvia genus, however, I came away with the impression that 80% of the genus is typically diploid, and that S. hispanica is diploid. While it is possible that different cultivars vary in this property, I did not find a paper presenting evidence of such diversity.
In response to Reviewer Two's questions surrounding Figure 3 and Figure 4, we decided to add more explanation to the figure captions for both figures. We also took this opportunity to import these two figures in SVG (vector) format rather than bitmaps. (We also added a TOC graphic, a feature we had previously overlooked.) We hope that Figures 3 and 4 will be improved by this effort.
Thank you for your detailed feedback!